# Is the Seasonal Variation in Frost Resistance and Plant Performance in Four Oak Species Affected by Changing Temperatures?

Maggie Preißer and Solveig Franziska Bucher *

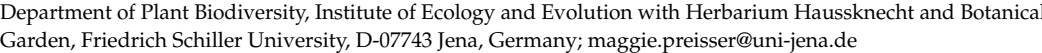

Department of Plant Biodiversity, Institute of Ecology and Evolution with Herbarium Haussknecht and Botanical Garden, Friedrich Schiller University, D-07743 Jena, Germany; maggie.preisser@uni-jena.de
* Correspondence: Solveig.franziska.bucher@uni-jena.de; Tel.: +49-3641-9-49986

**Abstract:** Research Highlights: We found seasonal variation in frost resistance (FR) and plant performance which were affected by growth temperature. This helps to better understand ecophysiological processes in the light of climate change. Background and Objectives: FR and photosynthesis are important plant characteristics that vary with the season. The aim of this study was to find out whether there is a seasonal variation in FR, photosynthetic $CO_2$ assimilation rates and leaf functional traits associated with performance such as specific leaf area (SLA), leaf dry matter content (LDMC), chlorophyll content, stomatal characteristics and leaf thickness in two evergreen and two deciduous species, and whether this is influenced by different temperature treatments. Additionally, the trade-off between FR and photosynthetic performance, and the influence of leaf functional traits was analyzed. By understanding these processes better, predicting species behavior concerning plant performance and its changes under varying climate regimes can be improved. *Materials and Methods:* 40 individuals of four oak species were measured weekly over the course of ten months with one half of the trees exposed to frost in winter and the other half protected in the green house. Two of these species were evergreen (*Quercus ilex* L., *Quercus rhysophylla* Weath.), and two were deciduous (*Quercus palustris* L., *Quercus rubra* L.). We measured FR, the maximum assimilation rate at light saturation under ambient $CO_2$ concentrations ($A_{max}$), chlorophyll fluorescence and the leaf functional traits SLA, LDMC, stomatal pore area index (SPI), chlorophyll content (Chl) and leaf thickness. Results: All parameters showed a significant species-specific seasonal variation. There was a difference in all traits investigated between evergreen and deciduous species and between the two temperature treatments. Individuals that were protected from frost in winter showed higher photosynthesis values as well as SLA and Chl, whereas individuals exposed to frost had overall higher FR, LDMC, SPI and leaf thickness. A trade-off between FR and SLA, rather than FR and photosynthetic performance was found.

**Keywords:** atLeaf; botanical garden; chlorophyll fluorescence; electrolyte leakage; gas exchange; PocketPEA



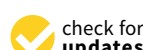

## 1. Introduction

Cold temperatures and frost limit the distribution and productivity of plants. Therefore, frost resistance (FR), the ability of a plant to sustain tissue functionality under cold stress alongside photosynthetic $CO_2$ assimilation rate is fundamental for plants to exist and persist in any given environment where frost potentially occurs [1–3]. Both processes can differ strongly between species and seasons, mostly influenced by photoperiod and climatic conditions such as temperature [1,4–6]. However, during the year plants need to balance between the maintenance of photosynthesis and the investment in FR as both are quite energy demanding. In this vein, a trade-off between these two traits is expected and has been reported previously [1,5,7].

FR can be captured via measuring the effective percentage of electrolyte leakage (Effective PEL, PEL$_{eff}$), which is an easy, cheap, and robust method for measuring high

numbers of plant individuals [7,8]. Capturing the photosynthetic performance can be achieved through various methods, such as determining chlorophyll a fluorescence via the JIP-Test [9–13]. There, both the maximum quantum yield of the photosystem II ($F_v/F_m$) indicating the efficiency of the photosystem II, and the absorption-based performance index ($PI_{abs}$) is assessed. In addition to that the gas exchange—i.e., the uptake of $CO_2$—can be measured directly [9,14–16]. The net $CO_2$ assimilation rate at saturation irradiance and ambient atmospheric $CO_2$ concentrations ($A_{max}$) can be used to assess the photosynthetic performance in a quick way and, using these values, the maximum carboxylation rate ($V_{cmax}$) can be calculated [9,14,15].

Research focuses increasingly on leaf functional traits capturing different processes of plant performance and adaptations to varying abiotic conditions [17–19]. Functional traits are thus expected to be linked to FR and photosynthetic performance. In our study, five traits were selected, namely the specific leaf area (SLA), the leaf dry matter content (LDMC), the chlorophyll content (Chl), the stomata pore area index (SPI) and leaf thickness. SLA (leaf area per leaf dry mass) is positively related to growth rates [18,20]. Due to the higher production of biomass and higher growth rates, the leaves with high SLA are short-lived [20–22], making them superior in resource-rich environments [23]. The SLA is also known to scale positively with the mass-based, light-saturated photosynthetic $CO_2$ assimilation rate [18]. LDMC (leaf dry mass per leaf fresh mass) has been shown to negatively correlate with SLA and growth rates and to indicate tougher, long-lived leaves, as it is a representative for the investment in structural compounds [18,20,23]. High LDMC is related to denser leaf tissues. Plants with high LDMC show slower decomposition rates and are more resistant to physical threats [18]. Chl can be used to determine the photosynthetic potential [24,25] and is connected to leaf nitrogen content [26]. A decrease in Chl is proven to indicate early senescence [27,28]. Stomata are the mediators of gas exchange managing the $CO_2$ uptake and water loss due to transpiration [29–31]. SPI considers the size of the stomata as well as the density and is known to affect gas exchange in plants [29,32]. In addition, the resource acquisition and resource usage strategy of the plants were measured through leaf thickness [33]. Thicker leaves show a greater amount of tissue per unit area [34], higher longevity [35] and absorb more light [36]. Negative relationships between the leaf thickness and the growth rates and between thickness and metabolic rates have been found [37,38].

FR, photosynthetic performance and leaf functional traits vary significantly in a species-specific manner throughout the year. Photosynthesis usually peaks in summer with maximum values in temperature and irradiation, and FR is greater in colder months in seasonal climates outside the tropics [1,3–5,39–42]. In this study, we analyzed how different abiotic conditions (plants with and without exposure to frost events) influence the previously described seasonal variability of FR, photosynthetic parameters and leaf functional traits (SLA, LDMC, SPI, chlorophyll content, leaf thickness) of four tree species (two evergreen, two deciduous) in order to assess the impact of changing temperatures on these fundamental trade-offs. These life forms differ in resource distribution, longevity of their leaves and their canopy duration in which photosynthesis can take place. FR is a major factor influencing performance also in deciduous trees, as they recover nutrients from senescing leaves before they are shed as these nutrients are stored and used in the next growth season and to protect their newly emerged leaves from late-frost events [2,3,5,43]. As leaves of deciduous trees are shed in winter their need to invest in FR is also different [43]. More precisely, we asked the following questions:

(1) Is there a difference in seasonal variation in FR, photosynthetic parameters, and leaf functional traits between the four species and between different temperature treatments?

(2) Is there a trade-off between FR and photosynthetic performance, and does it change under different temperature treatments?

(3) Are functional traits involved in this trade-off and can they be used to describe it?

By understanding the processes, predictions under changing climate regimes and in species behavior concerning the plant performance can be improved.

## 2. Materials and Methods

Four woody plant species, namely *Quercus ilex* L. and *Quercus rhysophylla* Weath. (evergreen species) as well as *Quercus palustris* L. and *Quercus rubra* L. (deciduous species) were selected for this experiment. Ten individuals of each species (around 2.5 m tall) were kept in the Botanical Garden of Jena, half of them experienced outside conditions, the other half was placed in a greenhouse inside the botanical garden without frost exposure, where temperature reached a minimum of 7 °C in winter in the nighttime and was set to a minimum of 9 °C during daytime (for temperatures in the botanical garden, please see Figure S1). Ventilation systems and windows helped to minimize temperature differences otherwise indoor and outdoor. Due to the size of the trees, no additional light was added in the greenhouse. The plants were watered as needed so that drought stress was excluded. All parameters were measured on a weekly basis from the 24th of September until the 3rd of June. During the winter period from December 17th until May 14th, measurements were carried out bi-weekly. Each week, two random sun-leaves were selected on each individual, as some measurements were destructive or left traces on the leaves. On both leaves, the chlorophyll fluorescence, Chl and leaf thickness were measured. One leaf was then used to measure FR, $A_{max}$, and SPI. The other leaf was used to measure SLA and LDMC. We chose to measure parameters with leaves still attached to the trees if possible, which could be done for $A_{max}$, chlorophyll fluorescence, Chl and thickness. Only after measuring these parameters the leaves were detached and brought to the lab to measure FR, SLA, LDMC and SPI. Leaves of the evergreen species could be measured more frequently, as the deciduous species fully shed their leaves by the 4th/19th of November (outside *Q. rubra/Q. palustris*) and the 26th of November (greenhouse), respectively. The new leaves of the deciduous species could be measured again starting the 27th of April or 12th of May (outside *Q. rubra/Q. palustris*) and the 14th of April (greenhouse), respectively.

### 2.1. Measuring of FR and Photosynthesic Parameters

The $PEL_{eff}$ was used to assess frost resistance. As described by [8], six 5 mm-diameter leaf fragments were cut out of every sample and were divided into two treatments (three replicates per treatment). One half was exposed to temperatures of $-32$ °C to simulate extreme frost (leading to values for $PEL_{frost}$), the other was kept at room temperature as a control ($PEL_{control}$). All samples were kept in their respective treatments for 14 h in complete darkness. After the treatment, the plant discs were removed from the freezer and allowed to come up to room temperature before conductivity was measured through a LAQUAtwin B-771 (HORIBA Instruments, Piscataway, NJ, USA). To obtain the data for maximum conductivity, the samples were boiled for 15 min at 100 °C. This caused a maximum membrane leakage and death of nearly 100% of the tissue cells. The PEL was then calculated as the quotient of the conductivity before and after boiling the samples for each treatment, thus assessing which percentage of cells were already destroyed through the treatment and was then multiplied by 100. $PEL_{eff}$ was then calculated as:

$$PEL_{eff} = PEL_{frost} - PEL_{control}.$$

High values of the effective percentage of electrolyte leakage indicate a low frost resistance. Negative values indicate that plants were less damaged in the freezer than in the control at room temperature.

Chlorophyll fluorescence was recorded using a portable continuous excitation time resolved Chl fluorimeter (PocketPEA, Hansatech Instruments, King's Lynn, UK). Every leaf was dark-adapted for 30 min using white leaf clips prior to the measurements of $F_v/F_m$ ($F_v$ being the variable fluorescence and $F_m$ the maximum value via the OJIP transient) and $PI_{abs}$ [44]. $F_v/F_m$ indicates the percentage of light quanta which are used in photochemistry, the $PI_{abs}$ gives species-specific information on plant performance which scales well with $A_{max}$ [9–13]. $A_{max}$ was measured using the Li-6400XT (LI-COR Bioscience, Lincoln, NE, USA). The block temperature was set to 20 °C to avoid the effect of changing temperatures throughout the year, irradiance was kept constant at 1500 µmol(photon) m$^{-2}$ s$^{-1}$ and the

$CO_2$ concentration stayed at 400 µL $L^{-1}$. The 'one-point method' was used to calculate $V_{cmax}$ thereof [14]. Negative values of $A_{max}$ and $V_{cmax}$ show that instead of an uptake of $CO_2$, we measured a release of $CO_2$, indicating that respiration was the dominant process over photosynthesis as in fact net-photosynthesis rates are measured via gas exchange.

### 2.2. Measuring the Leaf Functional Traits

SLA and LDMC were determined using a fine scale (ABJ, Kern & Sohn GmbH, Balingen, Germany) by measuring the dry and fresh mass of the leaf. The leaf area was recorded by scanning the fresh sample (CanoScan LiDE110, Canon, Tokyo, Japan). Both measurements were used to calculate the parameters using R [45] and the package LeafTraits (Bernhardt-Römermann, unpublished). Chl was measured in vivo using the atLeaf PLUS (atLeaf, Wilmington, DE, USA). Stomatal imprints from the abaxial and adaxial side were taken following the clear nail polish method [46]. The stomata of each side of the leaf were counted and the guard cell length was measured with a light microscope (AxioPlan, Zeiss, Jena, Germany). The SPI was calculated following [32]. None of the selected species showed stomata on the adaxial side of the leaf, which was checked on every leaf sample. Imprints could not be obtained for the leaves of *Q. ilex*. The thickness of each leaf was recorded via a digital caliper (Kunzer GmbH, Forstinning, Germany).

### 2.3. Statistical Analysis

To analyze whether parameters differed between species and the two temperature treatments, we performed analyses of variances (ANOVAs) followed by a Tukey's multiple comparisons of means test (Figure S2) using the parameters as dependent variables and the species as well as temperature treatment and the interaction thereof as explanatory variables. Mean values were only compared for the periods in which both life forms had measurable leaves.

Linear models were set up in order to test the seasonal variation of all recorded parameters. The parameters were used as dependent variables and day of the year (doy) as well as it's quadratic form $doy^2$ were used as the explanatory variables. The species identity, the temperature treatment and the interaction term between species:temperature treatment—as well as the interactions of species and temperature treatment, respectively with doy and $doy^2$—were all taken into account as a means to include species- and temperature treatment-specific responses. The full models were simplified via backward selection [47] to find the minimum adequate model. The interaction of $doy:doy^2$ was removed for all models considering the species were not observed over the full course of the year.

The trade-offs between FR and photosynthetic parameters ($F_v/F_m$, $PI_{abs}$, $A_{max}$ and $V_{cmax}$) were analyzed using linear models. FR was used as a dependent variable whereas the species identity, the temperature treatment and the photosynthetic parameter were used as independent variables. Additionally the interaction between species:temperature treatment, species:photosynthetic parameter and temperature treatment:photosynthetic parameter were included as independent variables. The models were simplified as described above.

In addition to that, the multivariate distribution of all parameters investigated was also analyzed using a principal component analysis (PCA). For every species in each temperature treatment, a confidence ellipsis was drawn in the plot to illustrate how species and treatments differed in a multivariate space. The statistical analyses were performed using R(version 4.0.3, Vienna, Austria) [45], the PCA was computed using the 'vegan' package [48] and displayed graphically using 'devtools' [49] and 'ggbiplot' [50].

## 3. Results

The measured parameters differed between species and between the temperature treatments. For LDMC, SPI and leaf thickness, individuals located outside had higher values than individuals located in the greenhouse (Table 1, Figure S2). The interaction species:temperature treatment was not significant in leaf thickness.

**Table 1.** Characterisation of four woody species with respect to their mean and range in measured trait values. Given is the mean value for each species as well as the range, i.e., minimum and maximum values measured, of frost resistance measured as effective percentage of electrolyte leakage ($PEL_{eff}$), net photosynthetic rate at saturating irradiance and ambient atmospheric $CO_2$ concentrations ($A_{max}$), maximum carboxylation rate ($V_{cmax}$), maximum quantum yield of PSII ($F_v/F_m$), absorption-based performance index ($PI_{abs}$), specific leaf area (SLA), leaf dry matter content (LDMC), stomatal pore area index (SPI), chlorophyll content (Chl) and leaf thickness (Thickness). The canopy duration indicates, during which period leaves were displayed on deciduous species. Significance is indicated as '***': $p < 0.001$.

| Species | Lifeform | Canopy Duration [d] | $PEL_{eff}$ [%] | $F_v/F_m$ | $PI_{abs}$ | $A_{max}$ [$\mu mol(CO_2)m^{-2}s^{-1}$] | $V_{cmax}$ [$\mu mol(CO_2)m^{-2}s^{-1}$] | SLA [$mm^2\,mg^{-1}$] | LDMC [$mg\,g^{-1}$] | Chl [$mg\,cm^{-2}$] | SPI | Thickness [mm] |
|---|---|---|---|---|---|---|---|---|---|---|---|---|
| *ANOVA* | | | $F_{4,794} = 38.7$ *** $R^2 = 0.16$ | $F_{7,647} = 15.9$ *** $R^2 = 0.15$ | $F_{4,650} = 30.2$ *** $R^2 = 0.16$ | $F_{7,764} = 10.7$ *** $R^2 = 0.09$ | $F_{7,752} = 8.0$ *** $R^2 = 0.07$ | $F_{7,748} = 289$ *** $R^2 = 0.73$ | $F_{7,688} = 170$ *** $R^2 = 0.63$ | $F_{7,781} = 155.2$ *** $R^2 = 0.58$ | $F_{5,516} = 69.6$ *** $R^2 = 0.40$ | $F_{4,794} = 299.2$ *** $R^2 = 0.60$ |
| *Q. ilex inside* | evergreen | | 27.4 [−13.0; 77.4] | 0.79 [0.46; 0.85] | 9.7 [0.03; 21.5] | 5.3 [−0.38; 13.6] | 9.4 [−0.34; 23.0] | 5.8 [1.1; 16.1] | 545.9 [358.1; 767.6] | 46.9 [18.5; 66.4] | - | 0.31 [0.21; 0.46] |
| *Q. ilex outside* | evergreen | | 23.2 [−19.8; 76.2] | 0.76 [0.57; 0.83] | 8.2 [1.3; 26.4] | 3.3 [−1.1; 9.9] | 6.3 [−3.5; 20.6] | 5.1 [2.6; 26.0] | 588.2 [491.5; 770.6] | 45.3 [19.2; 100.8] | - | 0.32 [0.20; 0.48] |
| *Q. rhysophylla inside* | evergreen | | 43.3 [−4.2; 118.7] | 0.80 [0.51; 0.84] | 8.0 [0.04; 23.9] | 4.2 [−0.26; 25.0] | 7.6 [−0.22; 26.2] | 11.7 [1.4; 37.5] | 456.2 [320.2; 579.3] | 38.0 [3.6; 52.1] | 37.6 [18.6; 56.1] | 0.23 [0.11; 0.37] |
| *Q. rhysophylla outside* | evergreen | | 34.6 [−1.6; 131.2] | 0.70 [0.34; 0.85] | 4.2 [0.12; 18.2] | 2.5 [−1.3; 9.6] | 5.1 [−3.9; 19.0] | 7.5 [3.3; 13.2] | 522.4 [332.2; 575.2] | 32.8 [13.7; 66.5] | 42.1 [7.6; 71.2] | 0.26 [0.17; 0.52] |
| *Q. palustris inside* | deciduous | 266–331; 105–155 | 53.3 [−9.5; 127.3] | 0.77 [0.25; 0.84] | 6.3 [0.31; 29.4] | 3.5 [−1.4; 14.8] | 6.5 [−3.5; 40.5] | 26.4 [14.0; 54.8] | 391.9 [269.0; 744.8] | 24.5 [6.9; 52.9] | 25.5 [11.4; 57.6] | 0.14 [0.06; 0.33] |
| *Q. palustris outside* | deciduous | 266–323; 133–155 | 47.8 [−33.7; 110.5] | 0.72 [0.39; 0.84] | 3.2 [0.02; 13.6] | 3.9 [−0.87; 12.4] | 6.6 [−4.9; 24.0] | 17.4 [11.6; 32.4] | 417.4 [226.4; 745.9] | 14.4 [4.1; 36.2] | 35.7 [21.0; 55.3] | 0.16 [0.05; 0.35] |
| *Q. rubra inside* | deciduous | 266–331; 105–155 | 41.1 [−19.9; 117.7] | 0.78 [0.23; 0.84] | 6.6 [0.16; 21.5] | 2.7 [−4.1; 12.1] | 5.0 [−5.1; 16.4] | 23.5 [6.8; 39.3] | 402.4 [274.0; 494.6] | 25.6 [3.9; 48.7] | 23.9 [12.0; 42.7] | 0.13 [0.05; 0.26] |
| *Q. rubra outside* | deciduous | 266–308; 118–155 | 36.0 [−27.4; 81.3] | 0.73 [0.37; 0.84] | 3.4 [0.02; 17.2] | 2.7 [−0.99; 12.2] | 5.0 [−1.6; 22.1] | 18.3 [11.4; 30.7] | 404.1 [277.6; 656.4] | 15.7 [4.3; 31.25] | 33.7 [17.4; 64.5] | 0.16 [0.08; 0.33] |

The opposite was true for $PEL_{eff}$, $F_v/F_m$, $PI_{abs}$, $A_{max}$ and $V_{cmax}$, SLA and Chl, where individuals located inside had higher values (Table 1, Figure S1). For $PEL_{eff}$ and $PI_{abs}$ the interaction species:temperature treatment was not significant.

### 3.1. Seasonal Variation and Species-Specific Responses

All parameters recorded showed a significant species-specific seasonal variation, yet not all of them were influenced by the temperature treatment in their seasonal variability.

In $PEL_{eff}$, $A_{max}$, $V_{cmax}$, LDMC and Chl we could not detect a significant interaction between doy:treatment or $doy^2$:treatment, indicating that temperature regimes did not change the seasonal variations in these parameters, just the overall values as indicated. $PEL_{eff}$ showed seasonal variation with higher FR in winter (Figure 1A; Table S1). $A_{max}$ and $V_{cmax}$ showed significant seasonal variation with deciduous species exhibiting decreasing values of the $A_{max}$ in autumn and increasing values in spring and evergreen species hardly showing any seasonal changes (Figure 1D,E; Table S1). LDMC showed a similar behavior in both, deciduous species, and evergreen species as for gas exchange measurements with a slightly more pronounced variation in evergreen species, which displayed maximum values in winter (Figure 1G; Table S1). Chl showed similar patterns for deciduous species as well, with evergreen species again reaching maximum values in winter (Figure 1H; Table S1).

On the other hand, $F_v/F_m$, $PI_{abs}$, SLA, SPI and leaf thickness differed between the two temperature treatments in their response to seasonal variation (Table S1). Individuals in the greenhouse showed higher values of $F_v/F_m$ and were also maintaining high values at the end of winter and in spring (Figure 1B; Table S1). Deciduous species showed increasing values of $PI_{abs}$ in autumn and decreasing values afterwards. Similar to $F_v/F_m$, individuals in the greenhouse showed higher values especially in spring (Figure 1C; Table S1). Increasing values of the SLA until the end of the year and decreasing values in spring were found for deciduous species. Similar to chlorophyll fluorescence, the difference between temperature treatments was most noticeable in early spring where individuals in the greenhouse had higher values (Figure 1F; Table S1). The deciduous individuals outside displayed decreasing values of the SPI until they had lost all their leaves, and increasing values in spring, whereas for evergreen individuals the variation in the greenhouse was much higher (Figure 1I; Table S1). A decrease in leaf thickness of the leaves could be observed in all species in autumn and winter. From the beginning of the new year the leaf thickness was increasing again. This variation was less pronounced in the greenhouse (Figure 1J; Table S1).

### 3.2. Trade-Off between Frost Resistance and Photosynthetic Performance

All relationships between FR and photosynthetic parameters were negative and did not differ in their slopes between the species or the temperature treatment, as the interactions of photosynthetic parameter:species as well as photosynthetic parameter:temperature treatment were not significant. The relationship between $PEL_{eff}$ and $F_v/F_m$ was significantly negative ($R^2 = 0.16$, $F_{5,543} = 20.55$, $p < 0.001$; Figure 2A) as was the relationship between $PEL_{eff}$ and $PI_{abs}$ ($R^2 = 0.15$, $F_{5,543} = 19.81$, $p < 0.001$; Figure 2B). Significant negative relationships between $PEL_{eff}$ and $A_{max}$ ($R^2 = 0.16$, $F_{5,543} = 20.55$, $p < 0.001$; Figure 2C) as well as between $PEL_{eff}$ and $V_{cmax}$ were also observed ($R^2 = 0.16$, $F_{5,543} = 20.2$, $p < 0.001$; Figure 2D). None of the described relationships were strong.

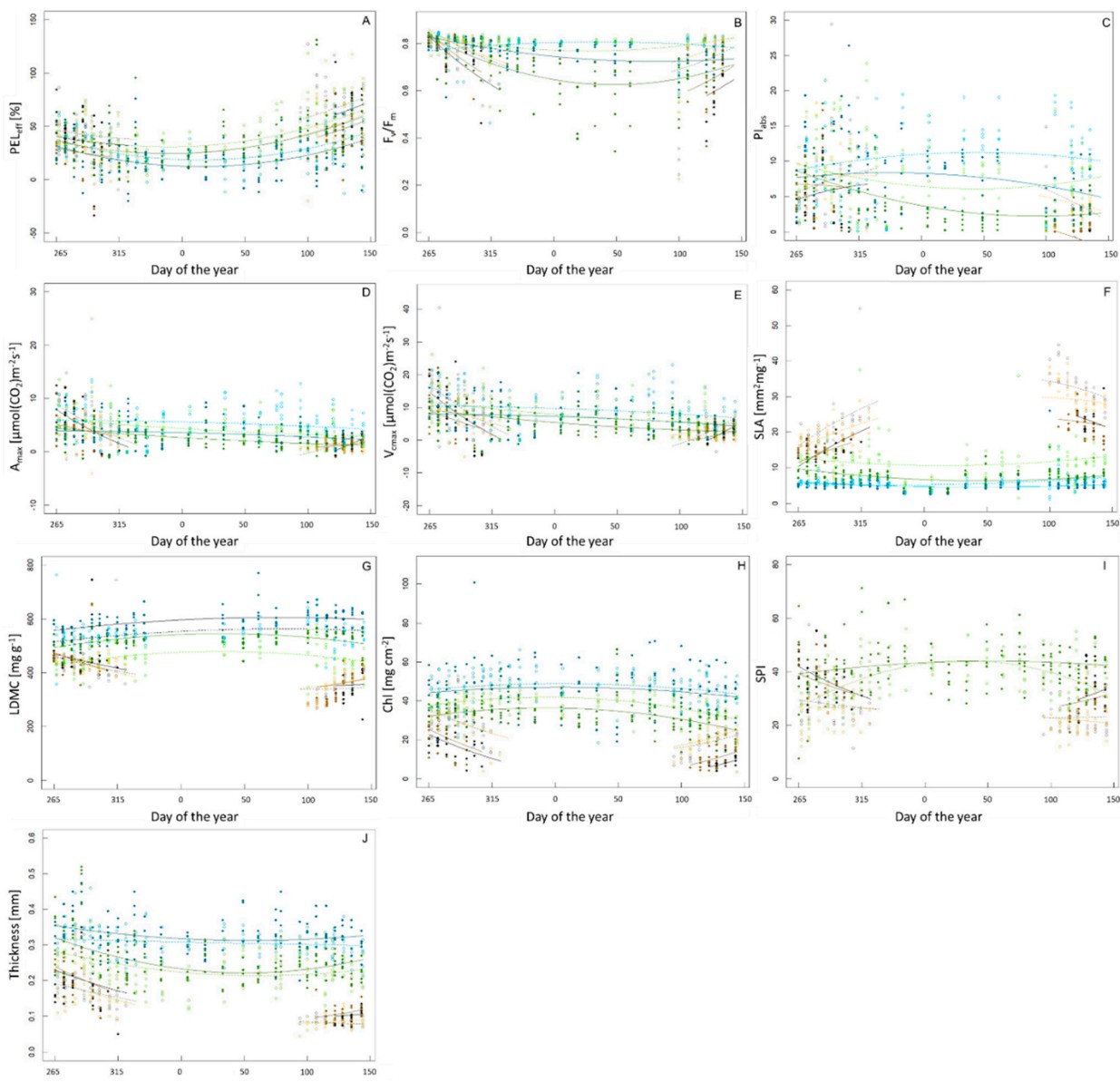

**Figure 1.** Seasonal variation of parameters recorded, namely (**A**) frost resistance as effective percentage of electrolyte leakage (PEL$_{eff}$), (**B**) efficiency of PSII (F$_v$/F$_m$), (**C**) performance index (PI$_{abs}$), (**D**) net CO$_2$ assimilation rate (A$_{max}$), (**E**) maximum carboxylation capacity (V$_{cmax}$), (**F**) specific leaf area (SLA), (**G**) leaf dry matter content (LDMC), (**H**) stomatal pore area index (SPI), (**I**) chlorophyll content (Chl) and (**J**) thickness of the leaf. Species are represented by color, the evergreen species *Q. ilex* is displayed in light blue (inside) and dark blue (outside). *Q. rhysophylla* was colored in light green (inside) and dark green (outside). The deciduous species *Q. palustris* is shown in light gray (inside) and dark gray (outside) and *Q. rubra* is displayed in yellow (inside) and brown (outside). Individuals and species situated in the greenhouse are represented by open circles and a dashed line, individuals and species that experienced outside conditions are represented by a diamond and a continuous line. The regression line was based on the minimum adequate models as described in the text.

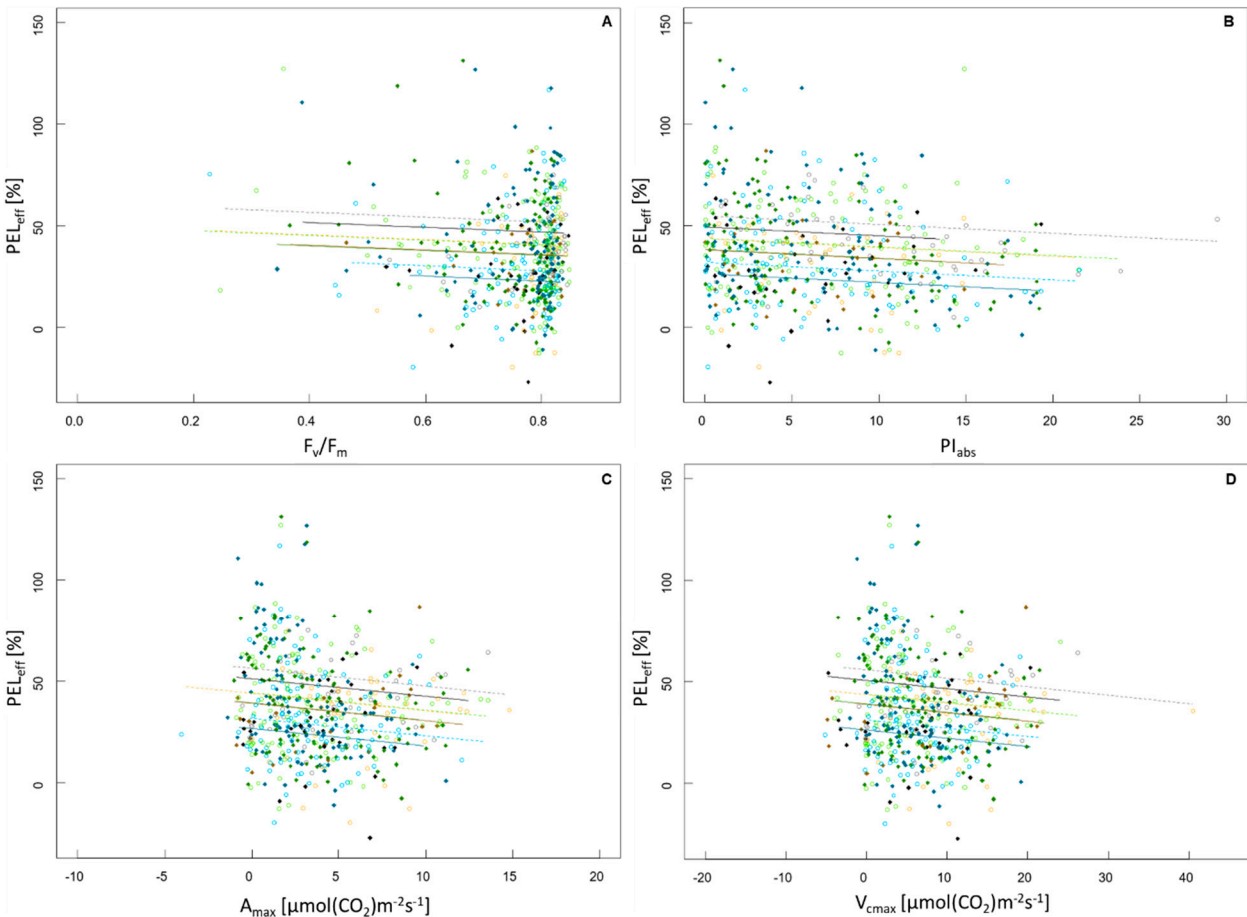

**Figure 2.** Trade-off between FR and photosynthetic performance represented by the relationship between the frost resistance as effective percentage of electrolyte leakage (PEL$_{eff}$) and (**A**) efficiency of PSII (F$_v$/F$_m$), (**B**) performance index (PI$_{abs}$), (**C**) net CO$_2$ assimilation rate (A$_{max}$), (**D**) maximum carboxylation capacity (V$_{cmax}$). Species are represented by color, the evergreen species *Q. ilex* is displayed in light blue (inside) and dark blue (outside). *Q. rhysophylla* was colored in light green (inside) and dark green (outside). The deciduous species *Q. palustris* is shown in light gray (inside) and dark gray (outside) and *Q. rubra* is displayed in yellow (inside) and brown (outside). Individuals and species situated in the greenhouse are represented by open circles and a dashed line, individuals and species that experienced outside conditions are represented by a diamond and a continuous line. The regression line was based on the minimum adequate models.

### 3.3. Effects of Traits on the Trade-Off between Resistance and Performance

The PCA (Figure 3) displays the distribution of all individuals within the multivariate leaf trait-space. The species were separated from each other as indicated by ellipses mainly on the first axis. The first axis (41.7% explained variance) primarily described the variation in the functional traits (SLA, LDMC, Chl, FR, and thickness as well as SPI (supplementary material Figure S3). Axis 2 (21.9% explained variance) was mainly associated with the photosynthetic performance parameters (F$_v$/F$_m$, PI$_{abs}$, A$_{max}$, V$_{cmax}$). There was a strong positive relationship of the PEL$_{eff}$ i.e., FR with SLA. Higher FR was related to higher LDMC, Chl, SPI and Thickness. The evergreen species are placed on the right-hand side and therefore have high values of FR, LDMC, Chl, SPI and leaf thickness, and lower values of SLA. Within each species, the individuals which were exposed to frost are all located more on the right hand side indicated by the ellipses and display the variation in traits along axis 1 in response to the temperature treatment and just a slight response along the axis 2 with slightly lower values of performance can be observed in the outside individuals.

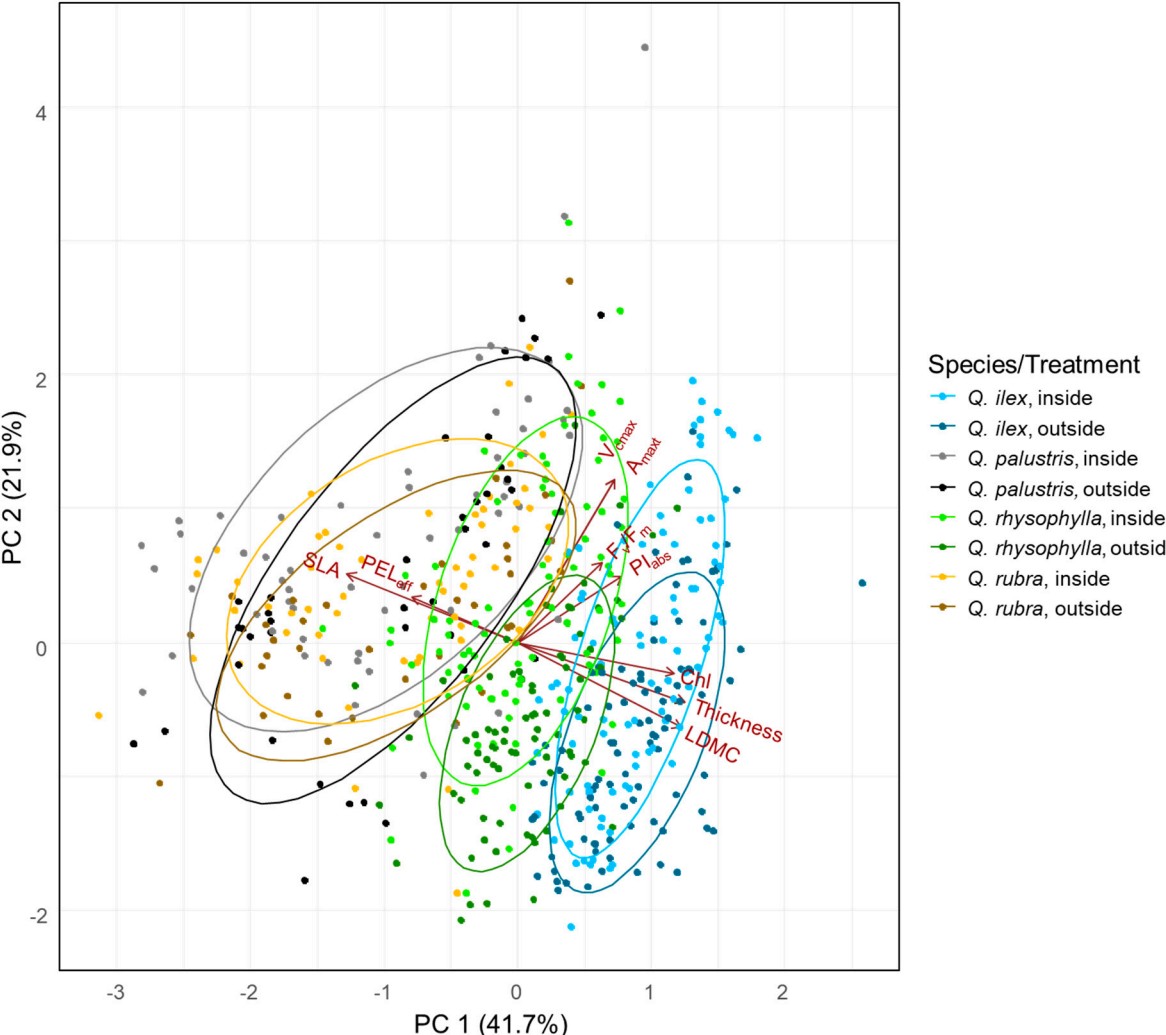

**Figure 3.** Principal component analysis of all traits, except SPI, and all species selected. Colors represent species and the associated treatment. The evergreen species *Q. ilex* is displayed in light blue (inside) and dark blue (outside). *Q. rhysophylla* was colored in light green (inside) and dark green (outside). The deciduous species *Q. palustris* is shown in light gray (inside) and dark gray (outside) and *Q. rubra* is displayed in yellow (inside) and brown (outside). Given is the frost resistance measured as effective percentage of electrolyte leakage ($PEL_{eff}$), the net $CO_2$ assimilation rate at saturating irradiance and ambient atmospheric $CO_2$ concentrations ($A_{max}$), the maximum carboxylation rate ($V_{cmax}$), the maximum quantum yield of PSII ($F_v/F_m$), the absorption based performance index ($PI_{abs}$), the specific leaf area (SLA), the leaf dry matter content (LDMC), the chlorophyll content (Chl) and leaf thickness (Thickness).

## 4. Discussion

All parameters recorded showed a significant difference between the selected species, between the two temperature treatments and showed a profound variation during the year, indicating a trade-off between growth and resistance. However, the PCA revealed that FR is rather associated with easy to measure traits like SLA, LDMC, Chl, leaf thickness and SPI than parameters directly assessing photosynthesis.

### 4.1. Influence of Temperature on Plant Traits

Plants located outside had higher values of SPI, leaf thickness and LDMC and slightly higher values in FR. Evergreen species showed higher mean values than deciduous species for FR, SPI and leaf thickness. A higher FR was expected for all species located in the outside treatment as they were exposed to frost, especially for the evergreen species which we could partially confirm. When pooled together, deciduous species had lower

FR as they shed their leaves during the cold period (Figure S4). Higher FR enables the plant to survive episodic frost events by preventing intracellular ice formation in their leaves [51,52]. For all species except *Q. ilex* (for $A_{max}$ and $V_{cmax}$ also *Q. rhysophylla*) all photosynthetic performance traits ($F_v/F_m$, $PI_{abs}$, $A_{max}$ and $V_{cmax}$) showed significant differences between the two temperature treatments and were often higher in the evergreen than in the deciduous species (Figure S4). As *Q. ilex* originates from the central-western Mediterranean basin [53], it might be possible that this species did not express a strong trade-off between FR and photosynthetic parameters overall because there is no need for this investment in its climate of origin. A higher photosynthetic performance for the other species might be possible considering the species in the greenhouse did not need to invest, as seen above, into FR which was also found by [1,5,7]. Also, Adams et al. [41] suggest that when temperatures are significantly below freezing no photosynthesis is possible. Again, except for *Q. ilex*, SLA was significantly different in the two treatments. A higher SLA in warmer temperatures was also detected by [54], yet might also be due to lower light availability in the greenhouse. Due to the positive relation of SLA with growth rates and mass-based, light-saturated photosynthetic rate [18,20] it might be suggested that the individuals kept inside were able to grow faster and maintain higher assimilation rates as shown above. The higher overall SLA values for deciduous species (Figure S4) may be due to the overall larger size of the deciduous leaves investigated and the need to reconstruct leaves in spring [5,6]. As shown by [37,38], growth rates decrease with increasing thickness of the photosynthetic structures, which we can confirm with our results using SLA as a proxy for growth rates. For the evergreen species, significant differences in LDMC values were found between the two temperature treatments which is also reflected in thickness. LDMC was higher in evergreen species (Figure S4). High LDMC values indicate a high resistance to physical hazards [18], which were also more pronounced outside as plants in the greenhouse were sheltered from wind. A higher LDMC also indicates tougher, longer-lived leaves as it is an indicator for the investment in structural compounds [18,20,23]. The higher Chl values of the species in the greenhouse could be attributed to absence of abiotic stress [55] or could be consequence of the lower light availability in the greenhouse. As for the photosynthetic performance parameters, evergreen species also had higher Chl values than the deciduous species (Figure S4), which may be due to the breakdown of chlorophyll during the course of senescence [9,56]. As Chl can be used to determine the photosynthetic potential [24,25], this suggests that evergreen species—and overall the species that were located inside the greenhouse—had a higher photosynthetic potential, which supports our findings for $F_v/F_m$ and $PI_{abs}$ and $A_{max}$ and $V_{cmax}$. All species (*Q. ilex* not included as no imprints could be taken) showed a significant difference in SPI between the temperature treatments. As SPI is known to have an impact on the gas exchange in plants [29,32], a higher SPI suggests a tighter regulation of gas exchange and higher potential conductivity for $CO_2$ and water vapor. For the species located outside, a stricter gas exchange regulation was needed due to the more rapid environmental changes compared to the greenhouse. As evergreen species had an overall higher leaf thickness, higher SPI could be found there (Figure S4). *Quercus rhysophylla* and *Q. rubra* had significant differences in thickness among the different treatments. Thicker leaves are associated with a higher longevity [35] and were associated with higher FR. It should furthermore be taken into consideration that sun leaves are known to be thicker [53,54] which could also explain the thicker leaves outside.

### 4.2. Seasonal Variation and Species-Specific Responses

Most parameters showed a strong seasonal variation which was influenced by the different temperature treatments in $F_v/F_m$, $PI_{abs}$, SLA, SPI and leaf thickness. For phanerophytes the main driving forces of FR are the photoperiod and temperature [2,3] which we could see in a hardening of the plants, yet less pronounced in the greenhouse. An increase in unpredictable episodic frost events [57–59] and earlier bud-break [60] are expected due to climate change and regionally rising temperatures. Therefore, species with a high FR—especially during the late spring and the beginning of autumn—would not be as damaged

through the suddenly low temperatures, ultimately benefitting by outbalancing the fewer resources that they put into photosynthesis. As already described by [42], $F_v/F_m$ was a good indicator of autumn senescence in deciduous species. The decrease of $F_v/F_m$ over the winter in evergreen species may be connected to the increase in FR and would confirm the hypothesized trade-off between FR and photosynthetic performance, as shown by [1,5,7]. $PI_{abs}$ showed similar responses as $F_v/F_m$. As for $F_v/F_m$, species located in the greenhouse showed an overall more linear seasonal trend than the species that experienced outside conditions, perhaps due to more constant temperature conditions, and they performed noticeably better especially in late winter and early spring, where they were sheltered from extreme weather conditions. Evergreen species were fairly constant in their $PI_{abs}$ values in winter and contradict the findings for the other photosynthetic performance parameters, which was also found by [1]. As shown by [1], similar patterns in seasonal variation between $A_{max}$ and $V_{cmax}$ were found. Light availability decreases during autumn, explaining lower photosynthetic performance for the evergreen species in this period of time, also found by [4,41,61]. The differences between higher seasonal variation in the fluorescence measurements and lower seasonal variation for the gas exchange measurements might arise due to the fact that only a few cell layers of palisade parenchyma cells are studied in Chl-alpha-fluorescence technique [62]. The temperature treatments did not influence the seasonal variation in photosynthesis indicating a control via photoperiod [63]. Deciduous species showed higher values of SLA at the beginning of the leaf out with decreasing values over the course of the year as they mature and fully develop, whereas evergreen species were fairly constant [1,6,63,64]. Plants in the greenhouse had higher values than individuals outside which was also reflected in chlorophyll fluorescence, indicating that due to the absence of frost plants could perform better mainly in early spring. High LDMC, being negatively correlated with SLA, is connected to the accumulation of non-structural carbohydrates [5,6,65] which accumulate, in the case of the evergreen species, especially in their overwintering leaves [1]. This might relate to the increasing values of FR at the end and beginning of the year. The deciduous species displayed an opposite trend with increasing values of the LDMC from the leaf-out and decreasing LDMC from the beginning of autumn confirming [64,65]. However, there was no impact of temperature treatment, meaning the driving force of seasonal variation may have been the photoperiod, as found for growth related parameters by [66]. The decreasing Chl values in autumn for deciduous species clearly indicated the beginning of senescence when chlorophyll breaks down [28]. Increasing Chl values at the beginning of the growing season were also observed by [67]. A slight opposite trend was recorded for the evergreen species, with slowly increasing Chl values during the winter period and slowly decreasing values at the beginning of summer. Since Chl is used to determine the photosynthetic potential [24,25], this corresponds well with the observations made for the photosynthetic performance parameters. The decreasing SPI values of deciduous species during autumn might be due to the approaching senescence and a loss of turgor, thus also the cell size. Highest value of SPI was observed in winter (also found by [68,69]), where there are frost events associated with drought [70], thus a tighter gas exchange control might be needed. As previously discussed by [29], stomatal size is genetically fixed as it is based on genome sizes so variations in SPI are more driven by changes in stomatal densities and not influenced by our temperature treatments which only affected winter temperatures. Because the amount of light absorbed by a leaf is dependent on leaf thickness [36], the increase in leaf thickness at the beginning of spring potentially enables leaves to acquire more light through additional layers of palisade parenchyma, ultimately being absorbed by the plant. Both, SPI and leaf thickness were more constant in the greenhouse (apart from SPI in *Q. rhysophylla*), probably due to the fact that abiotic conditions where more stable there. The outside species had more layers of palisade parenchyma as they receive higher radiation values, also reflected in higher LDMC, leading to a higher leaf thickness.

### 4.3. Effects of Traits on the Trade-Off between Resistance and Performance

A trade-off was found between FR and SLA (strongly negative relation), which would contribute to the plants' economics spectra [71–73] by finding a difference between performance and resistance. However, in a multivariate space, FR was much closer related to structural leaf traits than to photosynthesis traits. The evergreen species showed particularly high values of FR coupled with low SLA due to a conflict of resource investment [51,52]. The evergreen species were again found to have the highest FR and differed in their overall behavior compared to deciduous species. Further, all outside plants show higher FR and lower SLA values than the inside treatments. During winter, leaves of the evergreen species remained photosynthetically active but they seemed to invest the produced assimilates into carbohydrates, which are needed in FR rather than in growth [51]. This could also be due to different drivers as a high FR might be associated with lower temperatures, yet at the same time light levels were higher outside, inducing a lower SLA. Species with a higher LDMC showed higher FR and lower SLA, which was also found by [5]. This is also expected as LDMC is shown to negatively correlate with SLA [18,20,23] and might be linked to the accumulation of non-structural carbohydrates. High FR is related to high photosynthetic potential as Chl can be used to determine the photosynthetic potential [24,25] but not directly the photosynthetic parameters. This might indicate that FR shows a more direct trade-off to parameters of growth (SLA). On the other hand, it might be driven by leaf thickness and also be due to the measurement methodology of FR where thicker leaves take longer until all cells are destroyed, which might be by accident interpreted as high FR. The SPI is also known to be positively related to the net $CO_2$ assimilation rate and stomata conductance [29]. A high SPI, as found by [5], and low SLA can be indicators of high leaf thickness, which also correlates positively with FR in the PCA. In a study by [37], leaf thickness displayed a negative correlation to photosynthesis and respiration rates due to longer diffusion pathways or greater internal self-shading of chloroplasts [18], which could still give a hint of the former suspected trade-off between FR and the photosynthetic performance parameters, as found by [1].

### 5. Conclusions

The recorded parameters showed significant species-specific variation. There was a significant difference in traits between evergreen and deciduous species. Individuals that were kept inside the greenhouse showed higher values of $F_v/F_m$, $PI_{abs}$, $A_{max}$, $V_{cmax}$ as well as SLA and Chl, whereas individuals in outside conditions had overall higher FR, higher values of LDMC, SPI and thickness. The observed evergreen species showed higher mean values of each investigated trait except for SLA. A trade-off between FR and growth-rates (SLA) rather than FR and photosynthetic performance was found. The absence of a relation between FR and any of the photosynthetic performance parameters underlies this finding. Additionally, a positive relation between thickness, LDMC, SPI and the FR was found.

This study provides valuable insights into the seasonal variation and differences in FR, photosynthetic parameters, and leaf functional traits between evergreen and deciduous species, and between different temperature treatments. Further, it gives new insight into the trade-off between performance and resistance. Understanding the seasonal variation of these traits in other life forms besides phanerophytes and the differences between the two lifeforms with the differing temperature treatments—e.g., to fine-tune terrestrial biosphere models [74]—will increase their reliability by taking the trade-off between FR and plant growth into account.

**Supplementary Materials:** The following are available online at https://www.mdpi.com/1999-4907/12/3/369/s1, Table S1: Minimum adequate models describing the seasonal variation of all parameters depending on the four species and two temperature regimes. Figure S1: Daily minimum and maximum temperatures in the botanical garden. Figure S2: Graphical representation of differences in all parameters measured between the four species and the two temperature treatments. Figure S3: Principal component analysis of all traits selected without the inclusion of the species *Q*.

*ilex*. Figure S4: Graphical representation of differences in all parameters measured between the two lifeforms and the two temperature treatments.

**Author Contributions:** The research was conceptualized by S.F.B. and the data was collected by M.P. M.P. wrote the original draft of the manuscript, both authors discussed the statistical analysis and edited the manuscript. Both authors have read and agreed to the published version of the manuscript.

**Funding:** This research was funded by a research proposal granted to S.F.B. by the University of Jena (DRM/2018-05).

**Data Availability Statement:** After publications, this dataset will be submitted to the TRY database (http://www.try.org, (accessed on 16 March 2021)).

**Acknowledgments:** Special thanks to all employees of the Plant Biodiversity working group of the Institute of Ecology and Evolution with Herbarium Haussknecht and Botanical Garden, Friedrich Schiller University Jena, especially Laura Pabon for helping with the data acquisition and provision of lab facilities by Christine Römermann. We also thank the PhenObs project for providing temperature data of the botanical garden, especially Desiree Jakubka and Janin Naumann. Additional thanks to Timothy Brewer for language editing.

**Conflicts of Interest:** The authors declare no conflict of interest.

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
