# Peer review of "Is the Seasonal Variation in Frost Resistance and Plant Performance in Four Oak Species Affected by Changing Temperatures?"

_forests, doi:10.3390/f12030369_

Round 1

Reviewer 1 Report

Seasonal variation in frost resistance and plant performance in four oak species

The authors present a comprehensive study of frost resistance and associated leaf traits in several Quercus species. The study contains a great amount of measurements along different seasons and potentially interesting results. However, I would argue that as currently presented, the results are difficult to interpret by the reader as they are not presented in a straightforward manner. I am suggesting an important overhaul of the results before the article is ready for publication; as it is currently presented, it is very difficult to draw clear conclusions from the study.

Moreover, the authors should take care of using the ‘trade-off’ concept; see Grubb 2015 (‘Trade-offs in interspecific comparisons in plant ecology and how plants overcome proposed constraints’) for a really nice paper regarding this concept. I presume that the relationships found by the authors between FR and some structural leaf traits reflect a potential mechanistic link (a higher investment in leaves is required to sustain cold damage) rather than a true trade-off.

Here are some comments referred mainly to the methods and results presented:

Research Highlights – The sentence seems to broad for me; perhaps it should be narrowed to the main finding of the study.

Abstract

Line 22 – Delete ‘photosynthesis characteristics such as’, just state ‘maximum photosynthesis rate at light saturation under ambient CO2 concentrations (Asat)’ – Change ‘photosynthesis rate’ for ‘assimilation rate’. I would suggest using ‘An’ (net CO2 assimilation rate) here instead of ‘Asat’ as saturation could also refer to CO2 concentration.

Introduction

Line 36 – Clearly define ‘frost resistance’ for non-specialized readers. Do you refer to the ability to avoid low temperatures or to sustain tissue functionality under cold? Be more specific here.

Line 48-49 – Grammar: re-write this sentence as the enumeration of techniques as it stands is a little bit confusing.

Line 55-56 – Just as a curiosity... Could FR and photosynthesis be also included as leaf functional traits?

Lines 72-75 – Among the proposed traits, perhaps leaf thickness is the most spurious one, as it is a dimension related to LDMC. I would argue that this trait per se does not explain growth rates nor metabolic rates.

Material and Methods

Line 98 – Indicate the age of the individuals.

Line 99 – The conditions experienced by the plants need to be more detailed. You have to provide some kind of temperature monitoring data through the whole experimental period, clearly indicating the frost/extreme cold events experienced by the ‘outside’ plants. This part needs to be improved.

Line 117 – Is this temperature realistic? Do plants under the outside treatment experiment experience such low temperatures? Justify this value; most freezing experiments study temperatures from -20 to 0ºC.

Line 142 – I think when using these types of measures to estimate Chl content the authors should refer to it as ‘atLeaf value’ or ‘SPAD value’ to distinguish it from an actual measurement of Chl content, because the relationships between absorbance measured and Chl content are not always linear and species-specific. The authors could also perform a ‘calibration measurement’ using the atLeaf equipment and comparing the values for the studied species with chlorophyll content measurements. Otherwise, clearly specify that Chl content here is measured in vivo using leaf absorbance.  

Results

Line 179 – ‘within’?

Line 182 – use ‘temperatures’ through the manuscript, not ‘locations’ to refer to the frost treatments.

Lines 183-185 – This statement here is confusing; if Fr is measured as PELeff and one of the objectives of the study is to compare evergreen and deciduous species, then these two last sentences are contradictory (and repetitive!).

Figure 1 – This figure is difficult to interpret at first glance due to the small letters and excessive use of colours. It should be splitted into several figures to make each parameter easier to see. Or perhaps present these results as a table and highlight the differences for each parameter.

Overall, in the results section, you should be cautious when comparing deciduous vs. evergreen species, as you only present 2 species in each category (and therefore your analysis falls short to truly represent these two strategies). In order to avoid this problem, I would suggest to re-do the presentation of the first section of results as a table -or as more clearly presented graphs, and focus all the analysis as a Two-Way ANOVA having ‘temperature’ and ‘species’ as the main factors. Clearly indicate at the beginning of this part how do you take into account the measurements taken along the period of the experiment (line 155 in M&M is not enough – are these means along the periods? Ideally, you should include ‘time’ as a random factor!). Clearly present the ANOVA results (effect of each factor and their interaction).

The interpretation of the results along time is difficult due to the great amount of explanatory variables used in the models. In the Results text, please try to make it easier for the readers to extract the critical information – when the parameters differ between species and/or treatments.

Figure 2 – Again, the graphs presented here are difficult to interpret. Present, as clear as possible, the statistical analysis in the graph for each variable. Use colours and symbols for species/treatment (same for PCA).

Reviewer 2 Report

I was confused by the study aims; the title and the initial results (e.g. Fig 1) suggest that you are interested in inter-specific differences, but at many points in the text the main distinctions sought are between evergreen and summer-green species (e.g. L185 ).  This becomes an issue because the statistical results provided in the text show significant Species x Treatment interactions (df = 7), but the reader is left to try to pick through the boxplots which are arranged alphabetically rather than grouping by leaf habit.  And some of the claims are not supported: e.g. L190 – what species and treatment combinations are you talking about?  The control boxes for all species share the same Tukey identifier (Fig 1b).

You should consider running ANOVA tests for Leaf habit (evergreen/deciduous) x Treatment.

Again on the study aims (L88-89) I suggest rewording here.  Some of this seems self-evident.  e.g. is there seasonal variation between different temperature regimes? is there seasonal variation in frost resistance?  I think you need to justify your interest and approach in looking at frost resistance in deciduous species.  Many readers might ask ‘Why study seasonal variation in a deciduous plant?’  or ‘Why should a deciduous plant need to invest in FR?’  And the longitudinal study (Fig 2) combines leaves from two cohorts: autumn_Y1 and spring_Y2.

Some of the key traits e.g. PELeff, Asat and Vcmax are reported with negative values and this requires careful thought.  Should those findings be excluded from the analysis?  A negative PELeff implies that control plants have more frost resistance than those grown outside.  Is it sensible to talk about negative Vcmax?  It would make sense to screen the photosynthetic parameters for cases where stomatal conductance was heavily depressed.

The results section is highly repetitive with a paragraph going through each panel of each plot.  It would be much more interesting for the reader if you could provide a storyline and just pick out the most important findings.

Sect 3.1 – I could not make much sense of Figure 2; there were too many similar and overlapping lines.  And the model design was beyond me: how should I interpret treatment:doy2?

Sect 3.2 – Again very difficult to distinguish lines in Fig 3.  Fig 3a for example, I’m unsure how to interpret the test statistic; we are told that the relationship is defined by species and treatment, but that it is negative for all combinations.  Does the slope of the relationship differ depending on species x treatment?  The overall impression from the clouds of points in these four panels is that the relationships must be very weak and that is borne out by Figure 4 which shows the arrows for Asat and FR at right angles.

Minor points:

L21 - why not replace summer green with deciduous (passim)?

L29 - I disliked the adoption of SLA as a proxy for growth rate, better to say ‘photosynthetic performance or capacity’.  Photosynthate has many fates other than biomass production.

L43 - need some more background on the technique: what is the basis for comparison?  relative to what?

L57 – there is some redundancy here (borne out in the PCA in Fig 4); SLA (as the inverse of LMA) is the product of leaf density and thickness.  Also I think you need to provide the units here and a definition for the lay reader – not everyone knows what specific leaf area means.

L59 - why does a higher growth rate affect leaf longevity?

L72-73 – by definition.  I found it strange that this introductory paragraph has not invoked the leaf economics spectrum (ref 71) which captures most of these ideas.

L77-78 – need to qualify this by making it clear that you consider only ex-tropical environments.

L84-86 – sentence is too long to follow easily.  Try rewording.

L117-118 – was the frost exposure (-32C) also for 14 hours?

L121 - strange choice of wording: conductivity equates with leakage?  A bit more needed for the lay-reader e.g. conductivity is a measure for the amount of substance dissolved in a liquid.

L131 - again more for the lay reader: what are Fv and Fm, what do they tell us?  Also the SPI (L146) – some brief explanation of the technique required here, what are the units?

L133 - that is potentially quite a contrast for leaves grown outside and might constitute a shock.  In some instances there will be a big difference between growth temperature and measurement temperature.  Also these are excised leaves, kept at room temperature for 14 hours?  What about maintaining stomatal conductance?

L160 – replace location with ‘growing conditions (external or greenhouse)’.

L183 – the sentence here appears to contradict the one that follows.

L187 – what is meant by ‘within species’?

L378 - Reword - plants not species.  The growth treatment only produced a difference in FR for one species (Fig 1a).

L391 – show the citation differently e.g. Adams et al (41) show that ….

L418 - what is meant by aberrations?  Some problem with measurement or protocol?

L421 – what were the light conditions in the greenhouse? Natural daylight?

L463 - can deciduous species alter stomata numbers in a mature leaf?

L465 – turgor for turgesence.

L467-468 - how does that reconcile with the idea of seasonal variation above?

L479 – I think you mean structural traits?

Round 2

Reviewer 1 Report

All my major concerns and suggestions have been addressed and I think the manuscript is now improved. 

Author Response

Thank you for your effort and leading us to an improved manuscript!

Reviewer 2 Report

Still areas where the reader needs help: why is frost resistance a problem for deciduous plants?  what does the Fv/Fm ratio mean, how to interpret it?  Photosynthetic measures were taken for leaves still attached to the plant?

You need to explain negative values to the reader.  What do they show, why retain?  
